# The Reference Elevation Model of Antarctica

Ian M. Howat[1,2], Claire Porter[3], Benjamin E. Smith[4], Myoung-Jong Noh[1], Paul Morin[3]

[1]Byrd Polar & Climate Research Center, Ohio State University, Columbus, OH. USA
[2]School of Earth Sciences, Ohio State University, Columbus, OH. USA
[3]Polar Geospatial Center, University of Minnesota, St. Paul, MN. USA
[4]Polar Science Center, Applied Physics Laboratory, University of Washington, Seattle, WA. USA

*Correspondence to*: Ian M. Howat (ihowat@gmail.com)

**Abstract.** The Reference Elevation Model of Antarctica (REMA) is the first, continental scale Digital Elevation Model (DEM) at a resolution of less than 10 m. REMA is created from stereo-photogrammetry with submeter resolution, optical, commercial
satellite imagery. The higher spatial and radiometric resolutions of these imagery enable high quality surface extraction over the low-contrast ice sheet surface. The DEMs are registered to satellite radar and laser altimetry and are mosaicked to provide a continuous surface covering nearly [95%] the entire continent. The mosaic includes an error estimate and a time stamp, enabling change measurement. Typical elevation errors are less than 1 meter, as validated by the comparison to airborne laser altimetry. REMA provides a powerful new resource for Antarctic science and provides a proof of concept for generating high
resolution, accurate, repeat topography at continental scales.

## 1 Introduction

Ice sheet surface elevation is among the most fundamental datasets in glaciology. Often, investigations aimed at, for example, quantifying mass balance or ice flow modeling are limited by the spatial and temporal resolution and accuracy of surface elevation measurements. The polar regions have particularly poor topographic data due to their remoteness and the latitudinal
limits of global datasets, such as the Shuttle Radar Topography Mission (SRTM, limited to of 60°N and 56°S). For most of Antarctica, continuous grids of surface elevation, generally termed Digital Elevation Models (DEMs), have been limited to spatial resolutions of > 500 m and/or vertical errors reaching 10s of meters or more (e.g. DiMarzio et al., 2007; Griggs and Bamber, 2009; Cook et al., 2012; Fretwell et al., 2013; Helm et al., 2014; Slater et al., 2018). This limits their utility for geodetic applications, such as rectifying satellite imagery. Openly available, global DEMs, including the 30-m ASTER GDEM
(https://asterweb.jpl.nasa.gov/gdem.asp, last access: 13 Feb. 2019)   and recently released, 90-m TanDEM-X DEM (https://geoservice.dlr.de/web/dataguide/tdm90, last access: 13 Feb. 2019) have large errors over ice sheet interiors and, in the case of the latter, include a several meter bias due to penetration of the X band into firn (Wessel et al. 2018). Further, these DEMs do not have definitive time stamping, limiting their use for elevation change measurements. Since existing DEMs were mostly constructed from satellite ranging data (RADAR and LiDAR) for which errors increase with surface slope, errors tend
to be the largest in areas of more complex terrain, such as the coasts, mountain ranges and outlet glacier interiors (Bamber and Gomez-Dans, 2005). These areas, which include the Antarctic Peninsula and the Amundsen Sea outlet glaciers, are also the

areas where some of the largest changes in ice sheet dynamics and mass balance are occurring. Measurements of ice shelf thinning also require high precision because 1 meter of thinning equates to only ~0.1 meter of surface elevation change. Complex patterns of change, such as from ice shelf basal melting and subglacial hydrology are not often observable from current DEMs.

High precision elevations and elevation changes are obtainable from airborne laser altimetry but are only available along narrow (100s of m) swaths over limited areas of the ice sheet, and at infrequent intervals in time. While ICESat-2, launched September 15, 2018, will greatly increase the density and coverage of altimeter observations, it will not provide the continuous surface required for modeling and data processing applications. Further, a precise and time-stamped reference DEM will greatly benefit satellite altimeter missions by providing a validation surface, a basis for data filtering, and slope corrections for
radar ranges and offset ground tracks.

The objective of the Reference Elevation Model of Antarctica (REMA) project is to provide a continuous, time-stamped reference surface that is one- to two-orders of magnitude higher resolution than currently available (Fig. 1) and with absolute uncertainties of 1 m or less, depending on the availability of ground control, and relative uncertainties (i.e. precision) of decimeters. With REMA, therefore, any past or future point observation of elevation provides a measurement of elevation
change. Further, REMA may provide corrections for a wide range of remote sensing processing activities, such image orthorectification and interferometry, provide constraints for geodynamic and ice flow modeling, mapping of grounding lines, studies in surface processes and field logistics planning.

REMA was constructed from stereo-photogrammetric Digital Elevation Models (DEM) extracted from pairs of sub-meter resolution commercial satellite imagery and vertically registered to radar and laser altimetry data. Here we describe the source
datasets and the algorithms used to build REMA, as well as a validation of the final product using airborne altimetry from multiple sources.

## 2 Source Imagery and DEM Processing

REMA was constructed from stereoscopic imagery collected by four commercially-operated satellites: Worldview-1,2,3 and GeoEye-1, launched in 2007, 2009, 2014, and 2008, respectively (Table 1). These satellites are operated by DigitalGlobe Inc.
and their images are distributed via the Polar Geospatial Center (PGC) under a scientific use licensing agreement with the U.S. National Geospatial-Intelligence Agency (NGA). While the images are only available to U.S. federally-funded investigators, derived products, including DEMs, may be openly distributed. The pushbroom sensors aboard these polar orbiting satellites provide optical imagery with pixel ground resolutions of less than 0.5 m in the panchromatic band. Their camera pointing capabilities allow them to obtain overlapping images from different look angles, yielding convergence angles between image
pairs that are appropriate for stereo-photogrammetric DEM extraction. Using only the Rational Polynomial Coefficients (RPCs) derived from satellite positioning, these DEM's may have translational errors (biases) of several meters. These can be reduced through ground control registration to a point-to-point (relative) error of 20 cm (Noh and Howat, 2015; Shean et al., 2016), which is comparable to the uncertainty of airborne lidar. Importantly, unlike other common stereo-capable imagers,

such as from the Advanced Spaceborne Thermal Emission and Reflection Radiometer (ASTER), the high spatial and radiometric resolutions of these imagery enable high quality elevation extraction over low-contrast surfaces such as snow cover and ice sheet interiors/accumulation zones (Noh and Howat, 2015; Shean et al., 2016).

The satellite imagers targeted swaths of the ground surface each orbital pass, creating image strips that are up to 200 km long and between 11 and 17 km wide, depending on the sensor (Table 1). To ease data handling, the data provider divided each strip into approximately square subsets, or scenes, with ~10% overlap, prior to delivery. Pairs of strips covering the same ground area were selected for use as DEM stereo pairs if they had a convergence angle greater than 10° (Hasewaga et al., 2012), a difference in sun elevation angles of less than 10° and a time separation less than 10 days to reduce the likelihood of errors due to surface change, such as snowfall or ice motion. Pairs of images from the same sensor and different sensors were used. Through off-nadir camera pointing, we were able to successfully obtain DEMs over flat surfaces up to approximately 88° South. All REMA products are in polar stereographic projection, with a central meridian of 0° and standard latitude of -71°S and are referenced to WGS84 ellipsoid. A flow chart of the REMA workflow is provided in Fig. S1.

## 2.1 DEM Processing

DEMs were generated from scene pairs using the open-source and fully automated SETSM software package (Noh and Howat, 2017) on the Blue Waters Supercomputer at the National Center for Supercomputing Applications (NCSA). Prior to DEM processing, Worldview 1 and 2 images were destriped using the wv_correct function within the Ames Stereo Pipeline software package (Shean et al., 2016). SETSM DEMs commonly have artifacts at scene edges due to lack of constraint on Triangulated Irregular Network (TIN) generation and neighbor-based filtering at the scene boundaries. These artifacts appear as unrealistically high relief extending 10's to 100's of pixels from the scene edge. To detect and remove boundary artifacts, we computed the average slopes over square 21-pixel kernels. All pixels within a square 13-pixel radius of those with a mean slope greater than 1 were then removed. Enclosed gaps were then filled, so that only gaps touching the scene edge remain. Isolated clusters of less than 1000 pixels were then removed. A convex hull algorithm that includes concavity across data gaps was then applied to the remaining data to define the cropped scene boundary.

We additionally filtered each scene DEM for erroneous surfaces resulting from clouds or opaque shadows using the density of successful matches in the DEM extraction processes as given in the match tag file. We derive a match point density field by calculating the fraction of successful matches within square 21-pixel kernels. Pixels are then filtered if the match point density is below 0.9.

The filtered scene DEMs were then merged with adjoining scenes to form DEM strips comprising the overlapping area of the original stereopair image strips, performing three-dimensional coregistration using the iterative least-squares method of Nuth and Kaab (2011) and Levinson et al. (2013) and with distance-weighted averaging over the overlapping areas. Extensive erroneous surfaces due to, e.g., clouds or water bodies will cause errors in coregistration. Therefore, the scene was not merged if the root-mean-square of the residual differences in elevation between the overlapping area of the coregistered scenes was greater than 1 m. In this case, the strip was broken into separate segments and were treated as separate DEMs during the global mosaicking step described in Section 3. We note that the coregistration procedure may not provide correct horizontal offsets

in extremely flat, or uniformly sloping, terrain because the small range in slopes and aspects may not yield a confident regression. We could not identify such cases, however, suggesting that there is enough surface variation at these high resolutions (2-8 m) for the method to be successful.

From the archive of all imagery collected over the Antarctic continent as of July 2017, and with a cloud cover classification of 20% or less, we processed 79,362 individual strip pairs to create 187,585 DEM strip segments, with 66,401 of these covering West Antarctica and mountainous areas of East Antarctica processed to a resolution of 2m, and the remainder a resolution of 8 m (Fig. 1A). The lower resolution over regions of, generally, flat ice sheet surface was chosen to save computational costs. This equates to 122,567,288 km$^2$ of total coverage, including repeat coverage, and coverage of 13,987,485 km$^2$ (or 98%) of the continent, including islands laying greater than 60° south. The largest gap occurs over the "Pole Hole", south of approximately 88° south, with smaller gaps, mostly on occluded sides of mountains and in areas of persistent clouds such as the Antarctic Peninsula. These gaps will receive priority tasking in the future.

### 2.3 DEM Strip Quality Control and Registration

Hillshade representation images were generated for each DEM strip segment and these were visually inspected and classified based on visual quality (i.e. lack of erroneous surfaces due to clouds, shadows, etc). Such erroneous surfaces appear as chaotic textures in the hillshade image that contrast with the actual topography. DEMs were either accepted if no erroneous surfaces were identified in the hillshade image, manually edited to mask erroneous surfaces where errors were small and isolated, or rejected if errors were too extensive to be edited. Of the 187,585 strip DEM segments, 130,386 (69%) were visually inspected and classified. The remaining 31% of strips were not visually inspected because we switched from inspecting every strip to only inspecting strips needed for a single mosaic coverage part way through the quality control process. This resulted in fewer inspected strips for regions inspected after this change in procedure. Of the visually inspected strips, 43,550 (33%) passed quality control, with 19,971 (15%) requiring manual masking. In total, the 55,491,482 km$^2$ of quality-controlled DEMs cover an area of 13,567,969 km$^2$, or 95% of the continent (Fig. 1B).

All strips were vertically registered to altimetry point clouds obtained from Cryosat-2 radar and ICESat-1 Geoscience Laser Altimeter System (GLAS) campaign 2D (25 Nov. to 17 Dec. 2008). The ICESat-1 GLAS covers all areas of the Antarctica north of 86 degrees, with that limit, known as the "Pole Hole", due to orbital constraints. We use version 34 of the Level 2 GLAH12 altimetry data distributed by the National Snow and Ice Data Center (Zwally et al. 2014). The ground footprint of each laser shot has a diameter of approximately 70 m and an accuracy over flat surfaces of +/- 0.1 m with small variations due to snow surface properties (Shuman et al. 2006). Launched in April, 2010, Cryosat-2 carries the KU-band SAR/Interferometric Radar Altimeter (SIRAL) instrument with along- and across-track resolutions of 450 m and 1 km, respectively, in its higher resolution, interferometric (SARIn) mode. Cryosat-2 registration points were obtained from the Point Of Closest Arrival (POCA) locations in SARIn mode derived using a slope-threshold retracker (Gray et al, 2017). We use only the SARIn mode data and not the Low Resolution Mode (LRM) measurements because we did not feel confident that, over the scale of a DEM strip, the slope-driven error in LRM elevations would reliably average to zero. Each DEM strip was smoothed and down-sampled to a 32-m grid spacing, filtered to remove rough terrain, and then interpolated to the Cryosat-2 SARIn-mode point

cloud locations. For Cryosat-2 registrations, we estimated the linear temporal trend in the surface height from the time-series of all points within each DEM, so that each altimetric point measurement would provide an estimate of the surface height at the time of DEM acquisition. We did not apply a similar time-dependent correction to the ICESat-1 data because the time span between the altimetry measurements and the DEM was much larger. Further, we only use ICESat-1 data in the absence of

quality Cryosat-2 SARIn mode data, which is predominantly in the slow-flowing interior of the ice sheet where changes in surface elevation are expected to be less than the resolution of repeat surface height observations on sub-decadal timescales (e.g. Helm et al. 2014).

The median difference between the DEMs and the altimeter point clouds provides an estimate of the DEM's vertical bias. For Cryosat-2 data, only vertical bias corrections with a 1-sigma uncertainty of less than 0.1 m and residuals with a standard

deviation of less than 1 m were used in mosaicking. For ICESat-1, we impose a lower maximum threshold in the standard deviation of the residuals of 0.35 m because such strips were mostly used over the flatter interior terrain of Cryosat-2's LRM coverage. A total of 6,679,897 km$^2$ are covered by Cryosat-2 registered DEMs, or 29,901,958 km$^2$ including repeat coverage (Fig. 1C), with registered ICESat DEMs covering 4,897,600 km$^2$, including 8,739,128 km$^2$ of repeat coverage (Fig. 1D).

Strips with both Cryosat-2 and ICESat-1 registration within the bias correction uncertainty thresholds allow for an estimate of

the biases in Cryosat-2 height estimates due to the penetration of microwaves into the snow and firn layer (i.e. the penetration depth), or biases due to the retracking algorithm (i.e. where the retracker identifies a point on the leading edge of the waveform that does not correspond perfectly to the surface). Such biases are assumed negligible for the 1064 nm wavelength pulse of ICESat-1's laser altimeter and, therefore, the difference between the ICESat-1 and Cryosat-2 bias corrections should give an estimate of the Cryosat-2 bias. Fig. 2 plots the vertical bias corrections from ICESat-1 and Cryosat-2 for 227 strips for which

standard deviations of residuals were less than 0.25 cm. These strips were distributed across the entire area of Cryosat-2 SARIn coverage and, therefore, the mean difference between Cryosat-2 and ICESat-1 bias corrections should not be sensitive to local variability in surface elevation change over the period between the two missions. The mean difference between the two corrections is -0.39 ± 0.35 m. We expect the bias in the Cryosat-2 data to depend on surface density and surface slope (Wang and others, 2015), but we do not have a straightforward way of predicting the bias, and we did not find a clear spatial or

elevational dependence of the CS2-ICESat differences. Therefore, we added a uniform value of 0.39 m to the Cryosat-2-registered heights, regardless of the location of the strips and the surface type.

## 3 Mosaicking

Quality-controlled, strip DEMs were mosaicked into 100-km by 100-km tiles with a 1-km wide buffer on each side to enable coregistration and feathering between tiles. For each tile, strips with altimetry registration were added first, in order of

ascending vertical error, with a linear distance-weighted edge feather applied to the strip boundaries. The error value at each pixel is the standard error from the residuals of the registration to altimetry, and the date stamp is the day of DEM acquisition. The ± 0.35 m errors in bias for Cryosat-2 registered tiles were not included in this error estimate. In areas where edges of strips have been feathered, the error and date stamp are averaged with the same weighting as the elevation. Once all registered strips

were added, unregistered strips were added to fill gaps and are coregistered to the existing, registered data in the mosaic. Each quality-controlled, unregistered strip that overlaps a data gap was tested for the precision of three-dimensional coregistration, using the Nuth and Kaab (2011) algorithm, with the strip with the smallest coregistration error, defined as the root-mean-square of the elevation difference between the mosaic and the coregistered DEM, selected to fill the gap with the coregistration offset applied. Again, a distance weighted feathering was applied to smooth strip edges.

If Cryosat-2 registered data were available within a tile, those data were used, and any ICESat-1-registered data were ignored. If neither Cryosat-2 or ICESat registered data were available, the quality-controlled strip with the most coverage of the tile was added first and served as a relative reference. Unregistered strips were then coregistered to the mosaic and added as described above. Fig. 3 shows the distribution of tiles registered to Cryosat-2, ICESat-1 or alignment to neighbors.

Tiles around the edge of the ice sheet and within the zone of CryoSat-2 Synthetic Aperture Radar Interferometry (SARIn) mode collection, were mostly registered to contemporaneous Cryosat-2 altimetry, with the exception of coastal tiles with too little land surface or with extensive crevassing that prevented successful altimetry registration. Most of the interior tiles were registered to ICESat-1 and therefore have a nominal date stamp of late December 2008, although little or no secular surface elevation change is expected in these regions on sub-decadal time scales (e.g. Helm et al. 2014). Some tiles that were missing registration, and thus registered through alignment to neighboring tiles, are found around the Pole Hole and along a narrow zone in to its northeast. In most cases, the lack of registration was caused by a registration error larger than the thresholds defined in Section 2.3, likely because of the extreme off-nadir angles required for the satellites to acquire stereo imagery in the far south. Tile edges were feathered to smooth any offsets.

Finally, we applied a coastline mask using the British Antarctic Survey (BAS) land/ice classification polygons (https://add.data.bas.ac.uk/, last accessed 13 Feb. 2019). Since this coastline is of a lower resolution (10's to 100's of meters) and does not precisely match REMA in several areas, we masked as water all surfaces within 800 m of the coastline that were less than 2 m from the local mean sea level. Improving the delineation of the coastline is an objective of future versions of REMA.

The REMA mosaic includes both a vertical error estimate, based on altimetry registration, and coregistration for tiles aligned to neighbors, as described above, and grids of the day of data acquisition (Fig. 4). The 68[th] and 90[th] percentiles of errors are 0.63 and 1.00 m, respectively. Errors are highest in rougher terrain, such as the Antarctic Peninsula and Transantarctic Mountains. Higher errors also exist in zones of extensive crevassing along the coast and for tiles without control that are registered through alignment, and for which errors are thus propagated from the neighbors. The smallest errors are in the interior of the Cryosat-2 SARIn mask.

The mean date for REMA is 9 May 2015 with a standard deviation of 432 days. The mosaicking procedure resulted in no systematic distribution of date by acquisition time, but younger data tend to cover the higher latitudes, while older data cover areas of high science interest, as a result of long-term targeting. Our method of DEM registration to Cryosat-2 altimetry, described in Section 2.3, accounts for differences in time between the altimetry and DEM acquisitions, yielding temporal constraints on elevation for rapidly changing coasts and areas of fast flow. Even though much of the interior DEMs were registered to ICESat-1 data from late 2008, we retain the strip acquisition time in the date stamp as time-dependent changes in

these regions are expected to be small relative to the data error (e.g. Helm et al. 2014). Areas of local change, such as over subglacial lakes, should be small enough so as not to substantially effect tile registration. Caution, however should be used when assessing changes in tiles registered to ICESat-1. Tiles that are registered through neighbor alignment are given the weighted mean day of the data in the neighboring buffers.

We additionally filled gaps in the 100 m and coarser versions of the REMA mosaic using existing, lower resolution DEMs. For the Antarctic Peninsula area, we use the 100-m positing, edited ASTER GDEM mosaic by Cook et al. (2010). For the rest of the continent, we use Helm et al. (2014). We filled the mosaic by reprojecting and linearly regridding the lower-resolution DEM to REMA, differencing the regridded fill DEM with REMA in areas of data overlap, and then adjusting the fill DEM by the difference. The gaps are then filled with the adjusted fill DEM data. This process did cause artifacts in high relief areas

and where errors along the strip edges are propagated into the gaps by interpolation. These will be corrected or removed in future REMA versions as additional imagery are collected to fill gaps. Example hill shade representations of the REMA mosaic DEM that demonstrate the resolution are provided in Figs. S2, S3, S4 and S5.

    Two common artifacts in the REMA DEM are noisy surfaces due to opaque shadows, typically on the south sides of mountains, and repeated, horizontally offset surface features resulting from ice motion between stacked DEMs in the mosaic. Shadow

artifacts can occur in both the strip and mosaic DEMs and appear as rough surfaces over the area of the shadow. Shadows reduce the confidence of the stereopair matching algorithm within the DEM generation software, resulting in noisy surfaces. Examples of shadow artifacts are provided in Fig. S6. Ghosting only occurs in the mosaic, as it results from stacking multiple, overlapping DEMs, and is most commonly found on fast moving glaciers and ice shelves where crevasses and rifts advect with the ice. Ghosting artifacts appear as repeated, offset features that may fade in an out due to the feathering applied in the

mosaicking procedure. Examples of ghosting for ice shelf rifts are provided in Figs. S3 and S7.

**4 Comparison to Operation IceBridge Airborne LiDAR Altimetry**

    We provide an independent validation of the REMA strips DEMs and mosaic through comparison to airborne LiDAR altimetry acquired by the U.S. National Aeronautics and Space Administration's Operation IceBridge (OIB) between 2009 and 2017. Data from three different LiDAR systems are available: the Airborne Topographic Mapper (ATM), the Land, Vegetation and

Ice Sensor (LVIS) and the ICECAP laser altimeter system. The ATM is a conically scanning, 531 nm, 5 kHz LiDAR with a nominal footprint size of 1 m and a single shot accuracy of +/- 0.1 m (Martin et al. 2012). The LVIS system is a high-altitude, 1064 nm, 500 Hz scanning LiDAR with a 20-25 m footprint and a similar vertical accuracy as ATM (Hofton et al. 2008). The ICECAP laser altimeter operates at 905 nm with a footprint resolution of 25 m along track by 1 meter across track and an accuracy of 0.12 m (Young et al. 2014). All data were obtained from the National Snow and Ice Data Center (www.nsidc.org,

last access: Nov. 5, 2018). For ATM we used the Level 1B elevation data product, while we used Level 2 geolocated elevation products for the LVIS and ICECAP.

    All LiDAR data were provided in geographic coordinates referenced to the WGS84 ellipsoid and were converted to the REMA polar stereographic projection. We selected LiDAR data collected within 18 months of the REMA strip acquisition date or

mosaic date stamp. Strips were then three-dimensionally registered to each LiDAR point cloud, with the vertical residuals providing an estimate of precision. Histograms of the 68th and 90th percentiles of the absolute values of vertical residuals, or the Linear Error, LE, between each LiDAR system and the coregistered strips are shown in Fig. 5. The medians of the 68th percentiles are 0.44, 0.48, and 0.52 m for ATM, LVIS and ICECAP airborne lidars, respectively, and 0.84, 0.98 and 1.01 for

the 90th percentiles. These values are similar to those found by Shean et al. (2016) using DEMs created from the same imagery as REMA using ASP software, from Summit Camp, Greenland. They are, however, larger than the ~0.3 m found in comparisons between 2 m data and ATM over coastal Greenland, which is likely due to a combination of resolution and the larger, less rigorously quality controlled LiDAR and DEM datasets used here. Examination of outliers reveal that errors are often due to clouds and other errors in the various LiDAR datasets, as well as the DEMs. Thus, our data supports the finding

of Noh and Howat (2015) that the DEMs constructed from cloud free imagery with adequate illumination and appropriate base-height ratio, are of comparable internal accuracy (i.e. between locations on a single DEM) as available airborne lidar data. For tiles, which are registered to satellite altimetry through the mosaicking process described above, we linearly interpolate the REMA grid to the center coordinates of each overlapping LiDAR data point collected within one year of the REMA data, and differenced the interpolated REMA elevation from the LiDAR elevation. We then obtained the medians of the differences

of all points within each tile, as well as the 68th and 90th percentiles of their absolute values (termed the linear error, or LE68 and LE90 for the respective percentiles). Histograms of these values are shown in Fig. 6, and the medians and root-mean-square of the residuals are mapped in Figs. 7 and 8. REMA elevations are, on median, 0.06 and 0.47 m higher than ATM and LVIS, and 0.16 m lower than ICECAP elevations, while the LE68 values are 1.04, 1.19 and 0.77 m, and the LE90 values are 1.78, 1.74 and 1.25 m for ATM, LVIS and ICECAP respectively. The lower error values for the ICECAP data would be

expected due to the typically lower sloped terrain of East Antarctica, where these data are collected. We find no significant relationship, however, between slope and error. The median difference and root-mean-square error values mapped in Figs. 7 and 8 are largely consistent with those given by the Cryosat-2 and ICESat-1 registration errors in Fig. 4A, with the largest errors found over areas of crevassing and rifting on the coasts, in the high mountains of the Antarctic Peninsula and over tiles registered through alignment, such as around the Pole Hole. As with these strip comparisons, the comparison with tiles also

reveal errors in the LiDAR datasets, likely caused by clouds and aircraft positioning errors.

## 5 Dataset Attributes

The REMA datasets include all individual DEM strips (described Section 2.3) and the mosaic in 100 km by 100 km tiles (Section 3), all as 32-bit floating point raster files in GeoTiff format. The strip DEMs are either 2 or 8 m resolution, depending on region (Fig. 9) and include a metadata text file giving the version, projection and processing information. No ground control

or altimetry registration is applied to the strip DEMs in the current (Version 1) release. Version 1 includes 66,401, 2-m and 121,184, 8-m strip DEMs, totaling 45 TB uncompressed.

The mosaic is 8-m resolution everywhere and is registered to satellite altimetry data as described in Section 3. Each mosaic tile includes error estimate and date files, also in geoTiff format, as described above. The error file is 32-bit floating point

precision, whereas the date file is 16-bit integer precision in units of days since 1 January 2000. No void filling is applied to the 8-m tiles. Version 1 includes 1,524 mosaic tiles, totaling 1.0 TB uncompressed. In addition to the 8-m tiles, reduced-resolution, resampled versions are provided at 100-meter, 200-meter, and 1-km resolutions. The reduced-resolution datasets have an alternate filled version. Previews of the non-annotated and annotated, complete mosaic DEM hill shade representation maps provided by the PGC are shown in Figs. S8 and S9. Full resolution versions of the maps are available at http://maps.apps.pgc.umn.edu/id/2364 and http://maps.apps.pgc.umn.edu/id/2364 (last accessed 13 Feb. 2019), respectively.

**6 Conclusion**

Stereo-photogrammetry from high resolution commercial satellite imagery has enabled the first elevation mapping of nearly an entire continent at a horizontal resolution less than 10 m, and with a vertical error of less than 1 meter. The construction of REMA demonstrates the highly complementary characteristics of satellite altimetry, either from laser or radar, and stereo DEMs; altimetry provides highly accurate, but relatively sparse, control points to which the stereo DEMs provides a continuous surface of similar precision but lower accuracy. The combination of the two provides an effective method for maximizing resolution, coverage and accuracy.

Its geographic location, the flatness of the ice sheet and lack of vegetation all make Antarctica the easiest case for application of these methods. Polar orbiting satellites, with little competing demand for imagery provide the most abundant data at the poles. The flatness and lack of vegetation simplifies registration to satellite altimetry and ambiguities in the canopy versus ground height. These complications will need to be considered when expanding these methods to lower latitudes.

*Data availability.* All of the REMA products described above are openly available from the U.S. Polar Geospatial Center at www.pgc.umn.edu/data/rema. Imagery used to produce the REMA DEMs are available to U.S. federally funded researchers through the Polar Geospatial Center by request.

*Competing interests.* The authors declare that they have no conflict of interest.

*Acknowledgments.* This work was supported by U.S. National Science Foundation Office of Polar Programs Grants 1543501 to IMH and 1559691 to PM. High performance computing resources were provided by the National Center for Supercomputer Applications at the University of Illinois.

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

**Table 1. Specifications of Satellites and Imagery Used in REMA.**

| Satellite Name | Launch Date | Panchromatic Band Ground Sample Distance at Nadir (cm) | Swath Width at Nadir (km) |
|---|---|---|---|
| GeoEye-1 | 6 Sep. 2008 | 41 | 15.2 |
| WorldView-1 | 18 Sep. 2007 | 46 | 17.6 |
| WorldView-2 | 8 Oct. 2009 | 46 | 16.4 |
| WorldView-3 | 13 Aug. 2014 | 31 | 13.1 |

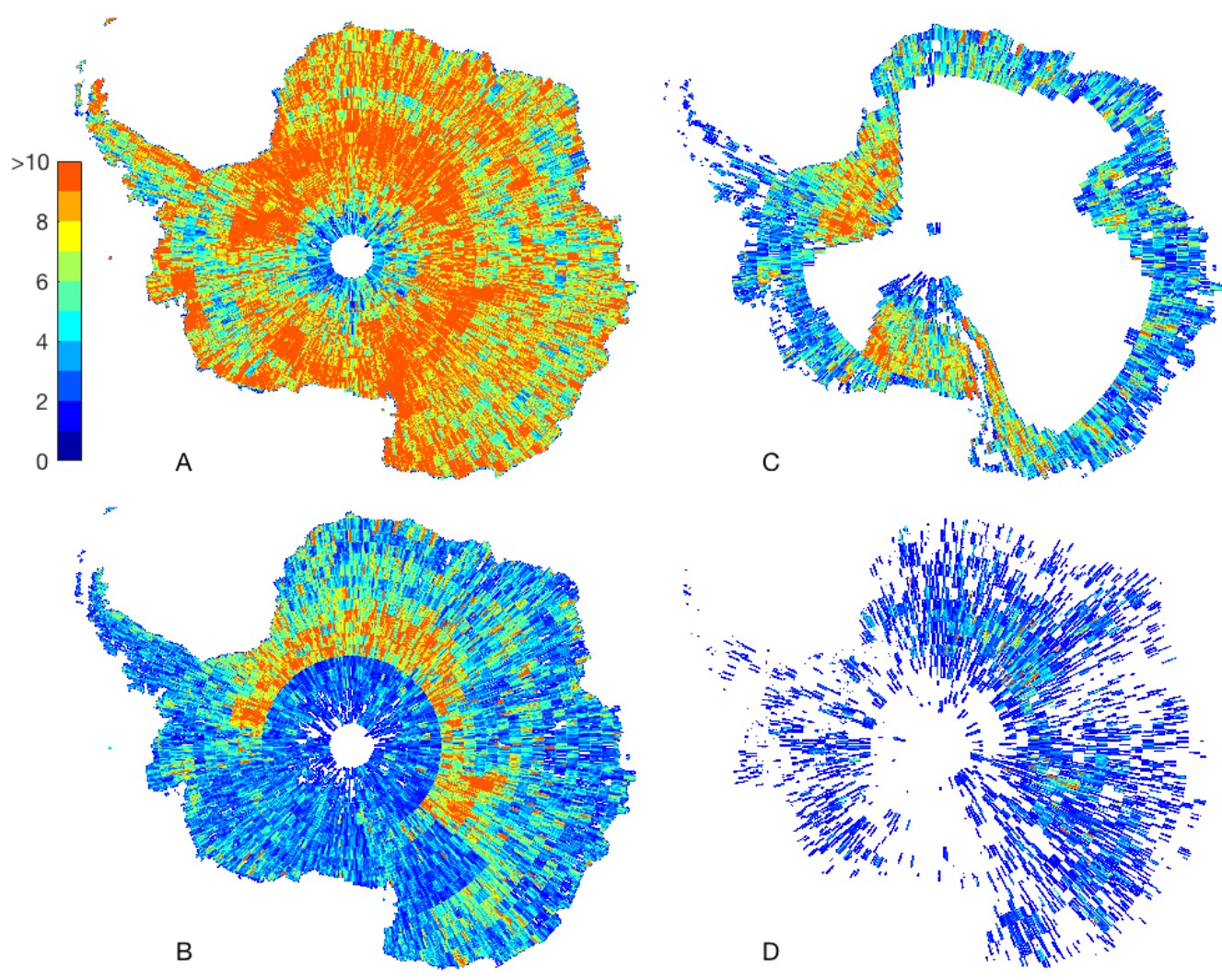

Figure 1: Maps of coverage of individual Digital Elevation Models (DEM) produced from stereoscopic submeter imagery for the REMA project, with color indicating the number of repeats, for (A) all data, (B) DEMs that passed visual quality inspection (note regional decrease in repeat coverage due to change in procedure), and quality-controlled DEMs with registrations within acceptable criteria from (C) Cryosat-2 and (D) ICESat GLAS campaign 2D.

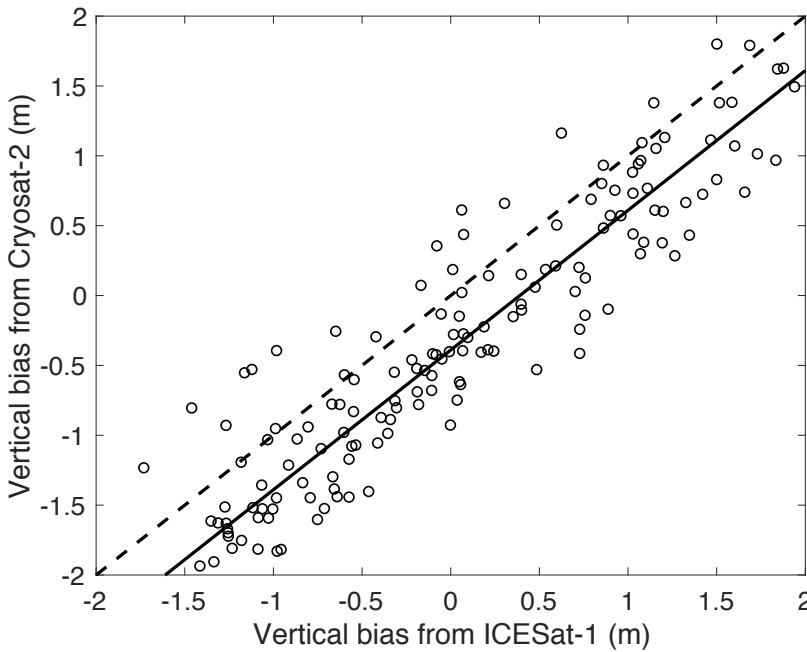

**Figure 2: Plot of vertical bias (i.e. the mean of residuals) between REMA strip DEMs and overlapping point clouds from ICESat-1 laser altimetry and Cryosat-2 radar altimetry. Only strips with standard deviations in residuals less than 0.25 m are plotted. Solid line is unity, shifted by the mean difference between biases (0.39 cm).**

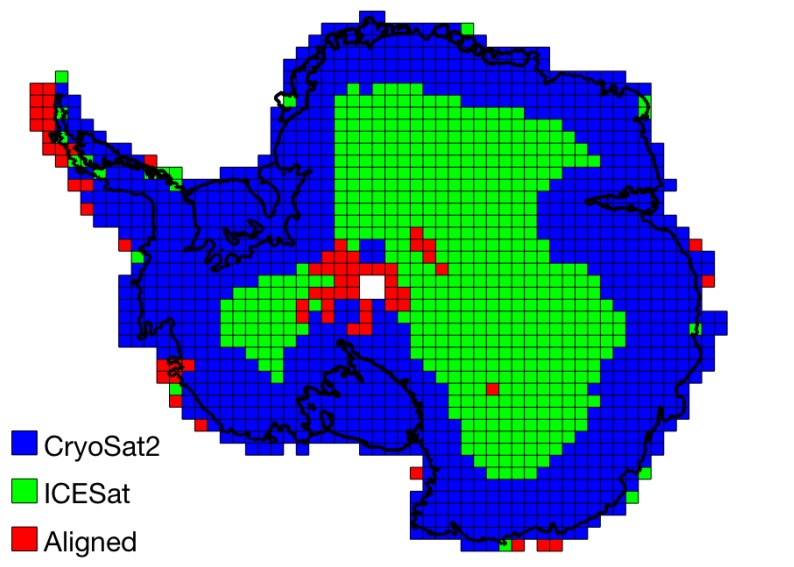

**Figure 3: Map of registration data source for each 100 km by 100 km REMA tile.**

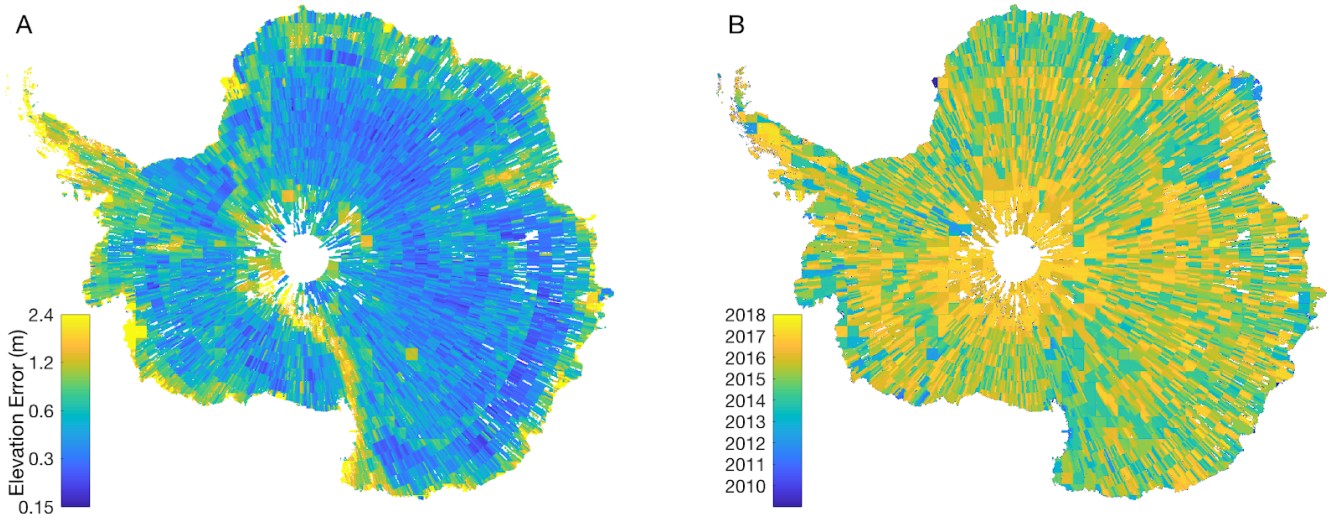

**Figure 4: Maps of REMA (A) elevation error, obtained from the root-mean-square of the differences in elevation between the DEM and altimetry data following registration, or the differences between co-registered DEMs in the case of alignment (note the logarithmic color scale), and (B) date stamp obtained from the date of image acquisition.**

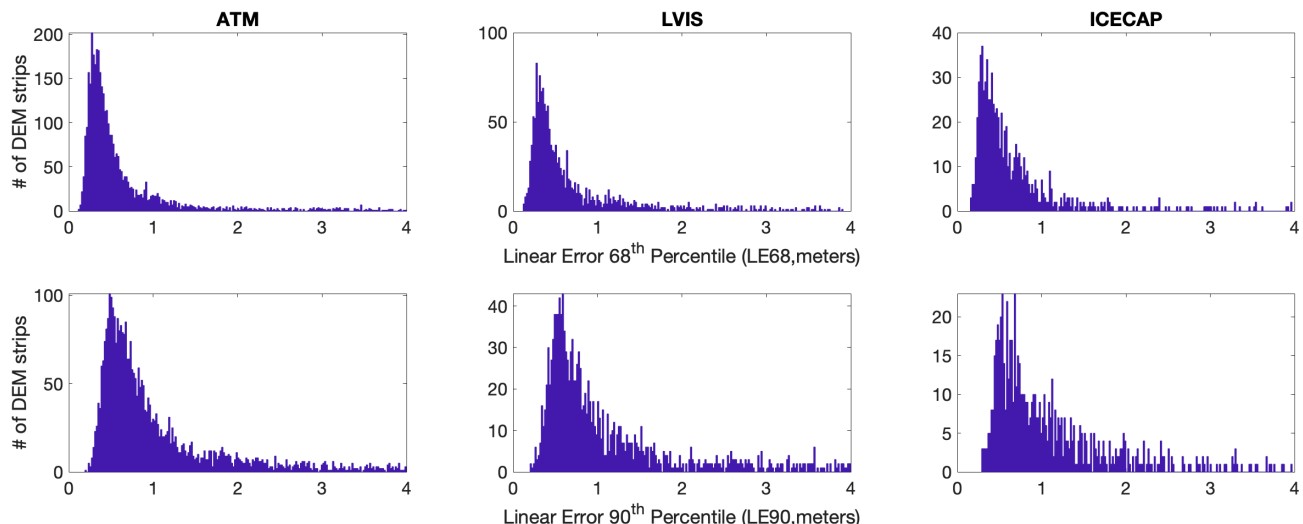

**Figure 5: Validation results for DEM strips. Histograms of 68th and 90th percentiles of the absolute value of vertical residuals, or linear error, between each of three Operation IceBridge LiDAR systems and REMA DEM strips after registration.**

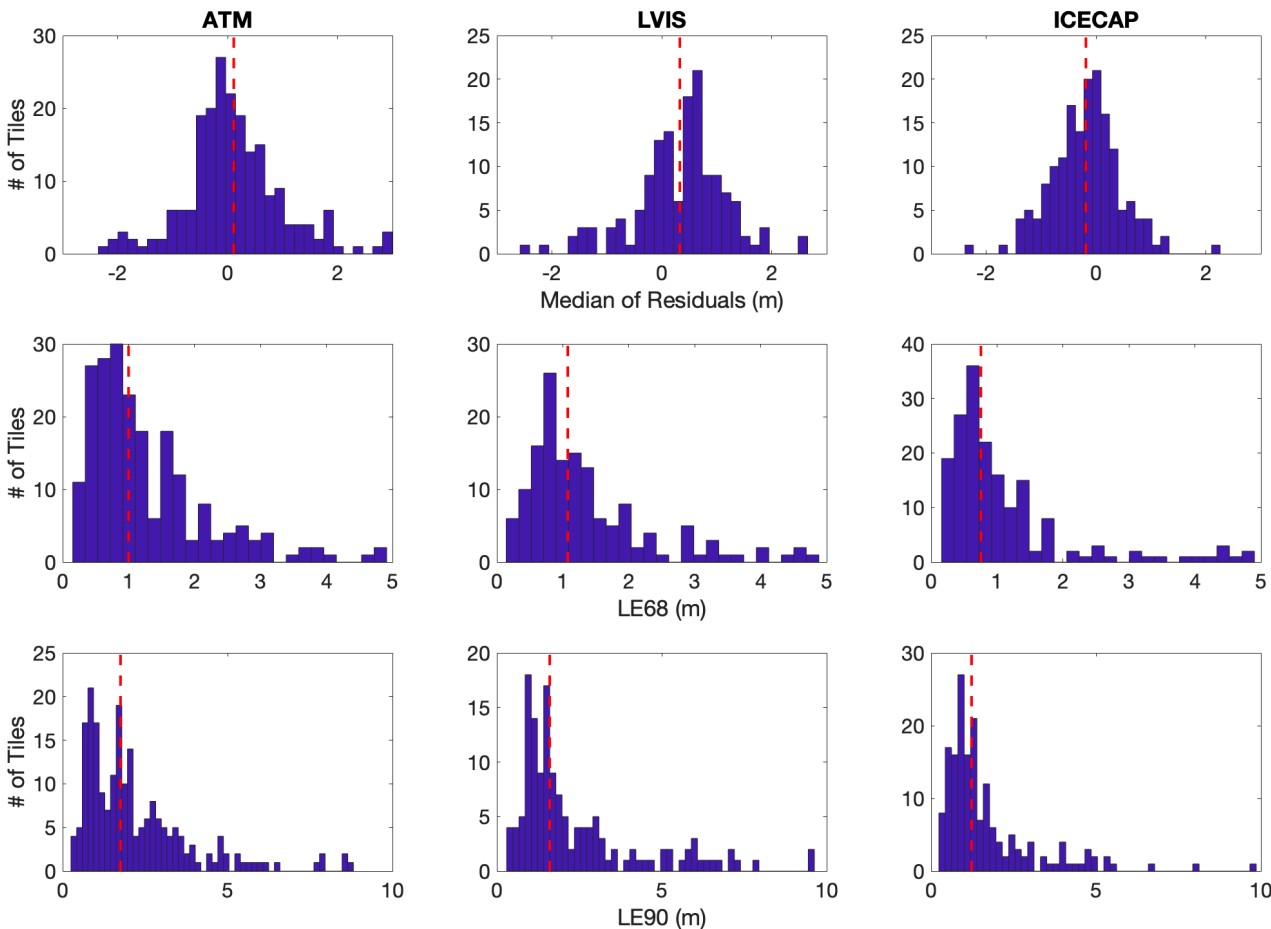

**Figure 6. Validation results for mosaic tiles. Histograms of the medians and linear errors at the 68th and 90th percentiles (LE68 and LE90) obtained from the differences between each REMA tile and the three NASA OIB airborne laser altimeters. The altimeter elevations are subtracted from the REMA elevations, so that a positive median of residuals (top plots) indicates the REMA surface is higher than the altimeter surface. Vertical red dashes are the median values.**

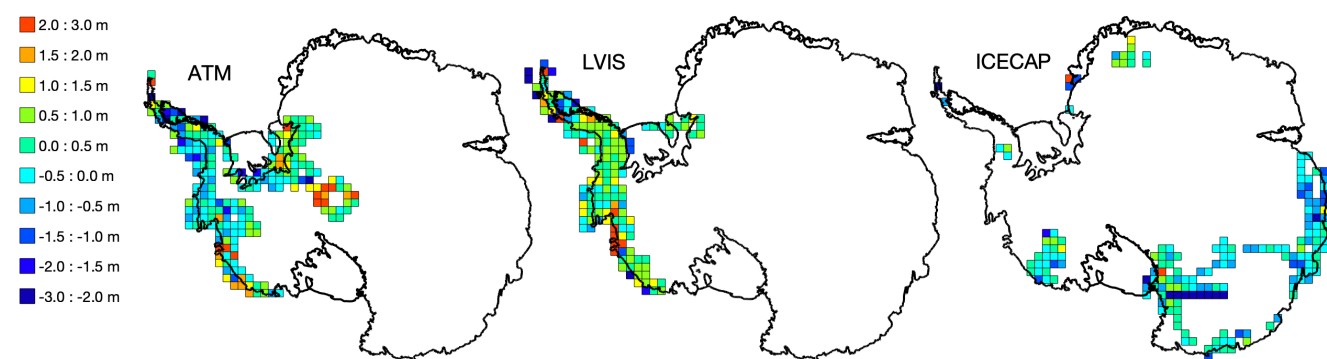

**Figure 7. Median of elevation differences by mosaic tile between REMA and each of the three NASA Operation IceBridge LiDAR systems. Only measurements collected less than one year apart are used.**

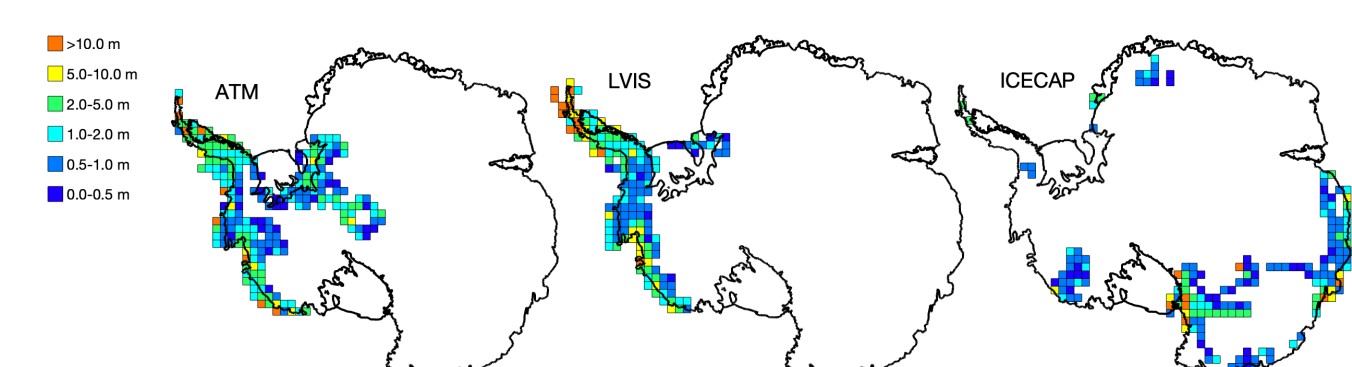

**Figure 8. Root-mean-square of point elevation differences by mosaic tile between REMA and each of the three NASA Operation IceBridge LiDAR systems. Only measurements collected less than one year apart are used.**

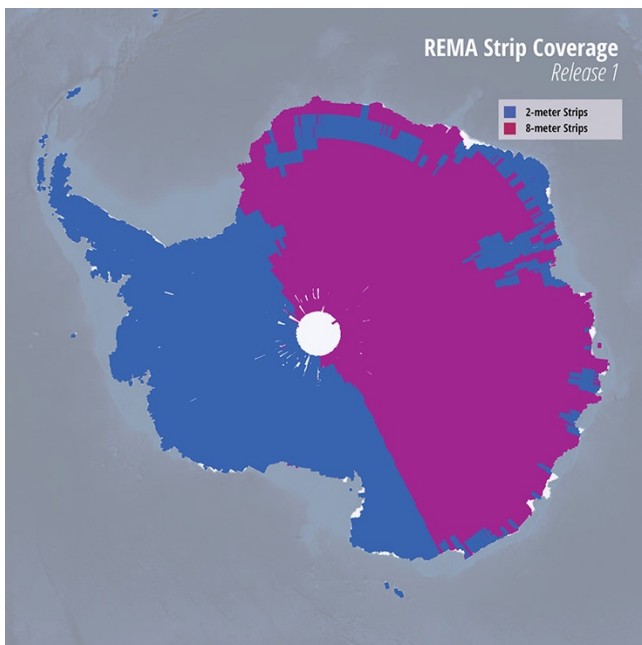

**Figure 9. Map showing coverage of 2- and 8-m resolution of DEM strips in the REMA version 1 release.**

