# Peer review of "The Reference Elevation Model of Antarctica"

_The Cryosphere, 2018_

## Referee Comment (RC1) · Anonymous Referee #1 · 11 Dec 2018

The paper "The reference elevation model of Antarctica" by Howat et al. presents the first digital elevation model (DEM) of the Antarctica continent at a spatial resolution better than 10 m.

The source data are provided by multi-spectral and high-resolution commercially-operated satellites. For this, neither the data nor the algorithms used for scene processing and mosaicking are novel and no relevant scientific contribution is provided in this sense. However, the resulting, openly available, data set represents a unique tool for the scientific community and a new standard for elevation measurements on the Antarctic continent.

Before publication, there are some - not critical - aspects that deserve to be taken into account. In general, the impression is that some part of the paper is somewhat short and superficial, and for this the authors should give some more insights and

explanations. These are detailed in the following:

- In Section 2.2 (Strip DEM Processing), the procedure for DEM registration by means of Cryosat-2 and ICESat-1 data is presented. Although well known to the community, some more technical details about these two sensors should be given, such as information on the SAR interferometer and type of sensor used (e.g. frequency, operation mode) for Cryosat as well as on the typical footprint and accuracy of ICESat.

- Again in Section 2.2 it is referred to the so-called "Pole Hole", the area around the South Pole which is not covered by any of the high-resolution source data. Why is that happening? For this, I guess the authors use then the ASTER DEM to fill the gaps. Did they try to check how does the seamless 90 m-resolution TanDEM-X DEM look like over the area?

- In Section 3, the authors discuss the filtering of water bodies by means of an external product, which has a lower resolution (this should be made explicit). For this, it is referred to a "buffering of the coastline by 800 m". What is meant with that, is it just a smoothing? Please explain.

- In Section 4, the validation of the product is presented. For this, the authors should clearly state with the help of basic but unambiguous formulas the parameters that they are considering for performance assessment (e.g. LE90, LE68, their absolute value), and that are plotted in the histograms (Fig. 5 and 6). This should be done for the sake of clarity and in order to avoid misunderstanding with the reader.

- In general, the authors should check the manuscript when they shortly refer to other datasets and products, and provide sufficient details for their easy "understanding". E.g. in Section 4, it is referred to a certain "qfit" data product for the ATM: here the paper would definitely benefit from a short description of this product.

- More examples of the resulting REMA product should be provided such as image zooms or detailed performance analyses (e.g. histograms) for selected DEM area

images, in order to give the user a feeling about the possible influences or problems when dealing with such a product (cloud cover? topography-related errors?).

- Please clearly state the difference between the histograms in Fig.5 and Fig.6: are the first related to all POINTS considered for validation, whereas the second is related to each 100 km x 100 km TILE?

- Considering the relative small amount of data and regions available for validation (according to Fig.8), is there the intention to extend it to larger areas of the continent? The authors should comment on this relevant aspect.

---

## Referee Comment (RC2) · Anonymous Referee #2 · 18 Dec 2018

General

This paper provides the description of the recently released REMA dataset for Antarctica. The dataset is revolutionary, providing high resolution continuous surface elevations for the entire continent. The amount of data processed alone is a remarkable accomplishment, and then combined with the heavy validation performed with ICE-Sat/Cryosat and ICEBridge will ensure the repeated and steady use of this dataset in the future for all Antarctic science. I congratulate the authors for making this possible, great job.

In general, the manuscript is well written, short and concise. In some ways, a bit too short, as there are many detailed processing steps that are somewhat brushed over which would make it difficult to reproduce the complicated merging and mosaicking of all individual DEMs. In light of this, I think it would have been useful for the authors

to also show/focus on some of the pitfalls of REMA showing a few examples of some of the more common problems and artifacts. This would help users of the dataset to easily spot artifacts when using REMA in their own research, especially those that are not accustomed to analyzing DEMs. In addition, It would have also been nice to see some advertising of the beauty of the dataset generated, for example by having a figure that exemplifies the precision using elevation profiles compared to ICESat and TanDEM-X, maybe one over some mountains, and another over the flat ice sheet with moderate topography. As of now, the figures all focus on the compilation of DEMs and their compiled accuracy, but no figures show the actual data at its natural resolution...

Here are a few comments towards the methods applied and described. I am particularly confused by the description and quantification of errors. By error (Fig 4a), do you mean the combined accuracy and precision (bias + random error). Maybe it would be useful to provide a final equation for how you attribute error to the individual tiles. This will be absolutely necessary for users to properly acknowledge and understand the abilities and constraints of the dataset. In terms of co-registration, it is often stated "coregistration residuals" which does not make much sense to me. Do you mean the elevation difference residuals after applying a 3D linear co-registration shift? Or do you mean the absolute vector of the co-registration shift. This needs to be clarified and used consistently through the text. Then, in terms of co-registration, the Nuth and Kääb (2011) approach is not solvable on flat terrain as the approach requires some slope and a bunch of aspects to solve properly. I imagine there is some consistent small scale topography of the ice sheet that was useful to use this approach. But in some areas where the distribution of slopes and aspects is limited, then the approach will probably only solve for a vertical bias. It would be useful to discuss this issue briefly, or at least mention it... Last, in terms of the correction inferred to derive from Cryosat-2 penetration, Since Cryosat is only used around perimeters in this study while ICESat is used in the interior, then, Do you think your spatial sampling biases the results here?

In summary, this manuscript provides a good description of a revolutionary dataset

for Antarctica, and will thereby be used and cited immensely. While there are limited major comments in this review, I hope the authors will find this useful to make their description even more transparent and clearer in order to help the plethora of users that will eventually integrate this dataset into their science.

Minor Comments P3, L29. I was confused by this header title. I suppose you are not combining individual images into strips, then processing DEMS from the combined strip images? Consider calling this section "Merging single scene DEM into along track strips" as this is what I inferred from this section. Please correct me if I am wrong. P4, L1: What is meant by "co-registration errors" ? Do you mean the magnitude of the vertical shift? How was this determined? P4, L11-17. What type of criteria is used in the visual inspection? And what is needed to pass quality control? Please provide additional details to make this process transparent, even though it is subjective to the inspector. P4, L31- . Was the sample used for this comparison spatially biased? Are all the points in this comparison located in one spot, or generally on the lower ice sheet. Additional details to clarify this would be helpful. Also, I wonder how the selective data approach (L26-27) by removing all strips that had a significant vertical bias (e.g. potential penetration '?) affected the interpretation of bias? If so, it could explain why you observe a "quasi" constant radar penetration estimate in Fig. 2, especially if all those scene residual statistics are spatially constrained on the continent. P5, L15. What is meant by "coregistration residual"? Do you mean the absolute magnitude of the co-registration vector? Did you apply the co-registration as well before filling the holes? P5, L19. Do you mean "absolute" reference? It would be relative reference if the strip was not aligned with ICESat/Cryosat, no? P5, L25-29. I am still confused about the "registration residuals"? If these are simply co-registration vectors, then I do not understand why they are used as residuals? For me, co-registration residuals would be the combined offsets between three or more datasets and subsequent triangulation of the co-registration vectors... (See Paul et. al. 2015) LAST: In such a massive undertaking for automated processing of DEMS and merging into a consistent product for the entire Antarctic Continent, would it not be useful to provide a flow diagram

showing the sequential processing, merging, and then mosaicking processing steps? I imagine that this procedure may happen again (repeat processing), from which others may learn significantly from the pipeline devised and implemented here. . .

References: Paul, F.; Bolch, T.; Kääb, Andreas; Nagler, T.F.; Nuth, Christopher; Scharrer, Killian; Shepherd, Andrew; Strozzi, T.; Ticconi, Francesca; Bhambri, Rakesh; Berthier, E.; Bevan, S; Gourmelen, Noel; Heid, Torborg; Jeong, Seongsu; Kunz, M.; Lauknes, Tom Rune; Luckmann, Adrian; Merryman, J.; Moholdt, G.; Muir, A; Neelmeijer, Julia; Rankl, Melanie; VanLooy, Jeffrey & Van Niel, Thomas (2015). The glaciers climate change initiative: Methods for creating glacier area, elevation change and velocity products.Remote Sensing of Environment. ISSN 0034-4257. 162, s 408- 426 . doi: 10.1016/j.rse.2013.07.043

---

## Referee Comment (RC3) · Bamber (Referee) · 19 Dec 2018

Review of Howat et al

**General comments**

The authors present a unique and incredibly rich data set that will have numerous applications, some mentioned in the m/s, but many that have yet to be thought of. Essentially, this paper is a brief summary of the DEM with a few statistics related to its accuracy. For a data set of this kind, that is to a large extent self explanatory in terms of its value and relevance, perhaps that's OK but there is useful and important information missing that would benefit the paper and any user of the product. Details of this are listed below but may not be exhaustive and I would encourage the authors to think carefully about what an end-user would benefit from here. A little more thought and perhaps even illustration of the potential applications of a data set of this unparalleled resolution would be welcome. How about some examples of shaded relief subsets where the full resolution can be seen over different types of terrain such as ice shelf rift areas, ice stream regions near the grounding line, and some examples of more rugged terrain around the Transantarctic Mountains and/or the peninsula. These would be helpful and instructive and make the m/s less dry.

**Specific comments**

1. Nowhere do you actually present a plot of the DEM itself. This seems like a pretty big oversight that is easily remedied. I suggest you include a supplementary figure at say 1:3,000,000 or a PDF/jpg version of the paper map that was distributed by PGC at AGU, which I note is available from the website. This can be a relatively large file and one version or other needs to accompany the paper.
2. Much of the "missing" information about the data set is available on the PGC website and includes, for example, the strip coverage at 2 and 8 m resolution. Strip DEM files sizes and file format. The fact that the DEM is 45 Tb is rather important for users to know as this present certain data handling and processing challenges.
3. P6, l9-16. I didn't really follow how the time stamp was generated for each strip: whether it was the date of the GCP acquisition or the image acquisition. If (as I suspect) it was the latter, then what did you do about any dh/dt trends that would offset your GCP elevations from the time stamp used? Much of the data in the interior seems to have a time stamp of ~2016-2018, almost a decade after the end of the ICESat mission.
4. Related to 3, I did not understand why you didn't use CS2 elevations from LRM data in the interior? The coverage is much better than ICESat and the accuracy comparable to SARIn mode data nearer the margins. Errors due to slope and effectively corrected in the interior. Requires explanation.
5. P3, l12. I think there is an error in the projection details provided. The std lat is most likely -71 degs and central meridian will be 0 degs not 71 degs. Otherwise it's all rotated with a non std pll.
6. P1, l23. Wrong reference to Bamber 2012. Should be Bamber, J. L., Gomez Dans, J. L., and Griggs, J. A. (2009), A new 1 km digital elevation model of the Antarctic derived from combined satellite radar and laser data. Part I: Data and methods, The Cryosphere 3(2), 101-111. Not the NSIDC URL.
7. P4, l31-35. The text in brackets could be better phrased. It's not picking a travel time but picking a point on the leading edge of the waveform that represents the surface. This point is a function of the retracking procedure. With a threshold retracker, the bias is a function of the choice of threshold. If the bias is really due to penetration (=> using a threshold that picks a point below the surface) then this will be a function of snowpack properties and, in particular, density. This may not have a clear relationship with elevation but should correlate with, say, surface density as estimated from an RCM. See, for example, Wang, F., Bamber, J. L., and Cheng, X. (2015),

Accuracy and Performance of CryoSat-2 SARIn Mode Data Over Antarctica, Geoscience and Remote Sensing Letters, IEEE, PP(99), 1-5, doi:10.1109/LGRS.2015.2411434.

8. P5, l4. Don't think "elevational" is a real word. Replace with elevation-related.

---

## Author Comment (AC1) · 4 Feb 2019

**Response to Referee 1**

**We thank the referee for her/his valuable suggestions. They have substantially improved the manuscript. Referee comments are in plain text below, with our responses in bold.**

The paper "The reference elevation model of Antarctica" by Howat et al. presents the first digital elevation model (DEM) of the Antarctica continent at a spatial resolution better than 10 m.

The source data are provided by multi-spectral and high-resolution commercially operated satellites. For this, neither the data nor the algorithms used for scene processing and mosaicking are novel and no relevant scientific contribution is provided in this sense. However, the resulting, openly available, data set represents a unique tool for the scientific community and a new standard for elevation measurements on the Antarctic continent.

Before publication, there are some - not critical - aspects that deserve to be taken into account. In general, the impression is that some part of the paper is somewhat short and superficial, and for this the authors should give some more insights and explanations. These are detailed in the following:

- In Section 2.2 (Strip DEM Processing), the procedure for DEM registration by means of Cryosat-2 and ICESat-1 data is presented. Although well known to the community, some more technical details about these two sensors should be given, such as information on the SAR interferometer and type of sensor used (e.g. frequency, operation mode) for Cryosat as well as on the typical footprint and accuracy of ICESat.

**We have added descriptions of both sensors to Section 2.2 as suggested.**

- Again in Section 2.2 it is referred to the so-called "Pole Hole", the area around the South Pole which is not covered by any of the high-resolution source data. Why is that happening? For this, I guess the authors use then the ASTER DEM to fill the gaps. Did they try to check how does the seamless 90 m-resolution TanDEM-X DEM look like over the area?

**The "Pole Holes" are due to orbital constraints and exists for most polar orbiters, not just our data. We have clarified this in the text. As discussed in section 3, we use the Helm et al. (2014) Cryosat-2 DEM for filling the mosaic, which uses interpolation at the pole hole. ASTER DEM has a similar pole hole and has very poor quality over the interior due to lack of optical contrast and its relatively low spatial and radiometric precision. The licensing of the Tandem-X DEM does not allow it to be included in our dataset for general release.**

- In Section 3, the authors discuss the filtering of water bodies by means of an external product, which has a lower resolution (this should be made explicit). For this, it is referred to a "buffering of the coastline by 800 m". What is meant with that, is it just a smoothing? Please explain.

**Resolution difference made explicit and revised sentence to read: "masked as water all surfaces within 800 m of the coastline that were less than 2 m from the local mean sea level."**

- In Section 4, the validation of the product is presented. For this, the authors should clearly state with the help of basic but unambiguous formulas the parameters that they are considering for performance assessment (e.g. LE90, LE68, their absolute value), and that are plotted in the histograms (Fig. 5 and 6). This should be done for the sake of clarity and in order to avoid misunderstanding with the reader.

**Clarified to read "We then obtained the medians of the differences of all points within each tile, as well as the 68th and 90th percentiles of their absolute values (termed the linear error, or LE68 and LE90 for the respective percentiles)."**

- In general, the authors should check the manuscript when they shortly refer to other datasets and products, and provide sufficient details for their easy "understanding". E.g. in Section 4, it is referred to a certain "qfit" data product for the ATM: here the paper would definitely benefit from a short description of this product.

**We remove "qfit" which is now obsolete, and have verified the clarity of other data descriptions.**

- More examples of the resulting REMA product should be provided such as image zooms or detailed performance analyses (e.g. histograms) for selected DEM area images, in order to give the user a feeling about the possible influences or problems when dealing with such a product (cloud cover? topography-related errors?).

**We have added the suggested examples to Supplementary Material and referenced them in the text.**

- Please clearly state the difference between the histograms in Fig.5 and Fig.6: are the first related to all POINTS considered for validation, whereas the second is related to each 100 km x 100 km TILE?

**Added statements clarifying this at the start of each caption.**

- Considering the relative small amount of data and regions available for validation (according to Fig.8), is there the intention to extend it to larger areas of the continent? The authors should comment on this relevant aspect.

We use all available NASA OIB data. As these validation data are collected via airplane, collection is heavily limited by logistics. We note that while the fractional area of coverage may appear small, Antarctica as whole is very big, and the total area of airborne lidar data used for validation is quite large (10's of thousands of km) and samples a wide range of terrains (mountains, ice shelves, plateaus, etc).

---

## Author Comment (AC2) · 4 Feb 2019

**Response to Referee 2**

**We thank the referee for her/his valuable suggestions. They have substantially improved the manuscript. Referee comments are in plain text below, with our responses in bold.**

General
This paper provides the description of the recently released REMA dataset for Antarctica. The dataset is revolutionary, providing high resolution continuous surface elevations for the entire continent. The amount of data processed alone is a remarkable accomplishment, and then combined with the heavy validation performed with ICESat/Cryosat and ICEBridge will ensure the repeated and steady use of this dataset in the future for all Antarctic science. I congratulate the authors for making this possible, great job.

**Thank you!**

In general, the manuscript is well written, short and concise. In some ways, a bit too short, as there are many detailed processing steps that are somewhat brushed over which would make it difficult to reproduce the complicated merging and mosaicking of all individual DEMs. In light of this, I think it would have been useful for the authors to also show/focus on some of the pitfalls of REMA showing a few examples of some of the more common problems and artifacts. This would help users of the dataset to easily spot artifacts when using REMA in their own research, especially those that are not accustomed to analyzing DEMs. In addition, It would have also been nice to see some advertising of the beauty of the dataset generated, for example by having a figure that exemplifies the precision using elevation profiles compared to ICESat and TanDEM-X, maybe one over some mountains, and another over the flat ice sheet with moderate topography. As of now, the figures all focus on the compilation of DEMs and their compiled accuracy, but no figures show the actual data at its natural resolution. . .

**We have added the suggested examples to the Supplementary Materials. We did not add the ICESat or TanDEM-X transects because those data ~10x lower resolution and do not show a "natural resolution" comparison. These data, as well as the airborne data, also have there own errors which makes such comparisons not straightforward - e.g. allocating which errors are REMA and which are the altimeter.**

Here are a few comments towards the methods applied and described.

I am particularly confused by the description and quantification of errors. By error (Fig 4a), do you mean the combined accuracy and precision (bias + random error). Maybe it would be useful to provide a final equation for how you attribute error to the individual tiles. This will be absolutely necessary for users to properly acknowledge and understand the abilities and constraints of the dataset.

**Expanded the the figure 4 caption to clarify this: "Figure 4: Maps of REMA (A) elevation error, obtained from the root-mean-square of the differences in elevation between the DEM and altimetry data following registration, or the differences between co-registered DEMs in the case of alignment (note the logarithmic color scale), and (B) date stamp obtained from the date of image acquisition."**

In terms of co-registration, it is often stated "coregistration residuals" which does not make much sense to me. Do you mean the elevation difference residuals after applying a 3D linear co-registration shift? Or do you mean the absolute vector of the co-registration shift. This needs to be clarified and used consistently through the text.

**This has been clarified in the text as described in the responses to the specific comments below.**

Then, in terms of co-registration, the Nuth and Kääb (2011) approach is not solvable on flat terrain as the approach requires some slope and a bunch of aspects to solve properly. I imagine there is some consistent small scale topography of the ice sheet that was useful to use this approach. But in some areas where the distribution of slopes and aspects is limited, then the approach will probably only solve for a vertical bias. It would be useful to discuss this issue briefly, or at least mention it. . .

**Added the statement to section 2.1: "We note that the coregistration procedure may not provide correct horizontal offsets in extremely flat, or uniformly sloping, terrain because the small range in slopes and aspects may not yield a confident regression. We could not identify such cases, however, suggesting that there is enough surface variation at these high resolutions (2-8 m) for the method to be successful."**

Last, in terms of the correction inferred to derive from Cryosat-2 penetration, Since Cryosat is only used around perimeters in this study while ICESat is used in the interior, then, Do you think your spatial sampling biases the results here?

**At the end of section 2.3: "We do not find a clear spatial or elevation-related dependence of this correction and, therefore, we applied the same correction to all strips regardless of location and surface type."**

In summary, this manuscript provides a good description of a revolutionary dataset for Antarctica, and will thereby be used and cited immensely. While there are limited major comments in this review, I hope the authors will find this useful to make their description even more transparent and clearer in order to help the plethora of users that will eventually integrate this dataset into their science.

Minor Comments

P3, L29. I was confused by this header title. I suppose you are not combining individual images into strips, then processing DEMS from the combined strip images? Consider calling this section "Merging single scene DEM into along track strips" as this is what I inferred from this section. Please correct me if I am wrong.

**The description of merging scenes into strips and the coverage of strips have now been merged into section 2.2. Section 2.3 is now titled "DEM Strip Quality Control and Registration"**

P4, L1: What is meant by "co-registration errors" ? Do you mean the magnitude of the vertical shift? How was this determined?

**Edited to read: "Extensive erroneous surfaces due to, e.g., clouds or water bodies will cause errors in coregistration. Therefore,  the scene was not merged if the root-mean-square of the residual differences in elevation between the overlapping area of the coregistered scenes was greater than 1 m. In this case, the strip was broken into separate segments and were treated as separate DEMs during the global mosaicking step described in Section 3."**

P4, L11-17. What type of criteria is used in the visual inspection? And what is needed to pass quality control? Please provide additional details to make this process transparent, even though it is subjective to the inspector.

**Edited to read: "Such erroneous surfaces appear as chaotic textures in the hillshade image that contrast with the actual topography. DEMs were either accepted if no erroneous surfaces were identified in the hillshade image, manually edited to mask erroneous surfaces where errors were small and isolated, or rejected if errors were to extensive to be edited."**

P4, L31- . Was the sample used for this comparison spatially biased? Are all the points in this comparison located in one spot, or generally on the lower ice sheet. Additional details to clarify this would be helpful. Also, I wonder how the selective data approach (L26-27) by removing all strips that had a significant vertical bias (e.g. potential penetration '?) affected the interpretation of bias? If so, it could explain why you observe a "quasi" constant radar penetration estimate in Fig. 2, especially if all those scene residual statistics are spatially constrained on the continent.

**We have added "These strips were distributed across the entire area of Cryosat-2 SARIn coverage."**

**The DEM selection criteria would not bias the offset between Icesat and Cryosat-2 (due to retracking and/or pentration) because the filter thresholds are applied to deviations in**

**residuals between the registered DEM and altimetry over each strip, not the mean of the residuals.**

P5, L15. What is meant by "coregistration residual"? Do you mean the absolute magnitude of the co-registration vector? Did you apply the co-registration as well before filling the holes?

**Edited to read: "Each quality-controlled, unregistered strip that overlaps a data gap was tested for the precision of three-dimensional coregistration, using the Nuth and Kaab (2011) algorithm, with the strip with the smallest coregistration error, defined as the root-mean-square of the elevation difference between the mosaic and the coregistered DEM, selected to fill the gap with the coregistration offset applied."**

P5, L19. Do you mean "absolute" reference? It would be relative reference if the strip was not aligned with ICESat/Cryosat, no?

**Correct and this is what is stated: "If neither Cryosat-2 or ICESat registered data were available, the quality-controlled strip with the most coverage of the tile was added first and served as a relative reference."**

P5, L25-29. I am still confused about the "registration residuals"? If these are simply co-registration vectors, then I do not understand why they are used as residuals? For me, co-registration residuals would be the combined offsets between three or more datasets and subsequent triangulation of the co-registration vectors... (See Paul et. al. 2015)

**"Residuals" should have been "error". The section now reads: "..the lack of registration was caused by a registration error larger than the thresholds defined in Section 2.3…"**

LAST: In such a massive undertaking for automated processing of DEMS and merging into a consistent product for the entire Antarctic Continent, would it not be useful to provide a flow diagram showing the sequential processing, merging, and then mosaicking processing steps? I imagine that this procedure may happen again (repeat processing), from which others may learn significantly from the pipeline devised and implemented here. . .

**A flow chart is now provided in the Supplementary Materials.**

References: Paul, F.; Bolch, T.; Kääb, Andreas; Nagler, T.F.; Nuth, Christopher; Scharrer, Killian; Shepherd, Andrew; Strozzi, T.; Ticconi, Francesca; Bhambri, Rakesh; Berthier, E.; Bevan, S; Gourmelen, Noel; Heid, Torborg; Jeong, Seongsu; Kunz, M.; Lauknes, Tom Rune; Luckmann, Adrian; Merryman, J.; Moholdt, G.; Muir, A; Neelmeijer, Julia; Rankl, Melanie; VanLooy, Jeffrey & Van Niel, Thomas (2015). The glaciers climate change initiative: Methods for creating glacier area, elevation change and velocity products.Remote Sensing of Environment. ISSN 0034-4257. 162, s 408- 426 . doi: 10.1016/j.rse.2013.07.043

---

## Author Comment (AC3) · 4 Feb 2019

**Response to Referee 3**

**We thank the referee for her/his valuable suggestions. They have substantially improved the manuscript. Referee comments are in plain text below, with our responses in bold.**

General comments
The authors present a unique and incredibly rich data set that will have numerous applications, some mentioned in the m/s, but many that have yet to be thought of. Essentially, this paper is a brief summary of the DEM with a few statistics related to its accuracy. For a data set of this kind, that is to a large extent self explanatory in terms of its value and relevance, perhaps that's OK but there is useful and important information missing that would benefit the paper and any user of the product. Details of this are listed below but may not be exhaustive and I would encourage the authors to think carefully about what an end-user would benefit from here. A little more thought and perhaps even illustration of the potential applications of a data set of this unparalleled resolution would be welcome. How about some examples of shaded relief subsets where the full resolution can be seen over different types of terrain such as ice shelf rift areas, ice stream regions near the grounding line, and some examples of more rugged terrain around the Transantarctic Mountains and/or the peninsula. These would be helpful and instructive and make the m/s less dry.

**Four sample images over varying terrains have been provided in the Supplementary Material.**

Specific comments
1. Nowhere do you actually present a plot of the DEM itself. This seems like a pretty big oversight that is easily remedied. I suggest you include a supplementary figure at say 1:3,000,000 or a PDF/jpg version of the paper map that was distributed by PGC at AGU, which I note is available from the website. This can be a relatively large file and one version or other needs to accompany the paper.

**The map (blank and labelled versions) are now included in the supplementary material and are referenced in the text.**

2. Much of the "missing" information about the data set is available on the PGC website and includes, for example, the strip coverage at 2 and 8 m resolution. Strip DEM files sizes and file format. The fact that the DEM is 45 Tb is rather important for users to know as this present certain data handling and processing challenges.

**We have added a new section (5 Dataset Attributes) that summarizes the characteristics of the dataset, including formats, sizes, etc., and include a new figure (9) that maps the 2m and 8m coverage.**

3. P6, l9-16. I didn't really follow how the time stamp was generated for each strip: whether it

was the date of the GCP acquisition or the image acquisition. If (as I suspect) it was the latter, then what did you do about any dh/dt trends that would offset your GCP elevations from the time stamp used? Much of the data in the interior seems to have a time stamp of ~2016-2018, almost a decade after the end of the ICESat mission.

**This is now clarified to read: "Our method of DEM registration to Cryosat-2 altimetry, described in Section 2.3, accounts for differences in time between the altimetry and DEM acquisitions, yielding temporal constraints on elevation for rapidly changing coasts and areas of fast flow. Even though much of the interior DEMs were registered to ICESat-1 data from late 2008, we retain the strip acquisition time in the date stamp as time-dependent changes in these regions are expected to be small relative to the data error. Areas of local change, such as over subglacial lakes, should be small enough so as not to substantially effect tile registration. Caution, however should be used when assessing changes in tiles registered to ICESat-1. Tiles that are registered through neighbor alignment are given the weighted mean day of the data in the neighboring buffers."**

4. Related to 3, I did not understand why you didn't use CS2 elevations from LRM data in the interior? The coverage is much better than ICESat and the accuracy comparable to SARIn mode data nearer the margins. Errors due to slope and effectively corrected in the interior. Requires explanation.

**We did not use the LRM measurements because we did not feel confident that, over the 10's of km scale of a DEM strip, the slope-driven error in LRM elevations would reliably average to zero. Although it may be possible to make a correction for this effect, and it may not result in a significant error over the flat parts of the interior, we felt that the errors due the time differences between the Worldview data and ICESat data were easier to understand than errors in the LRM dataset.**

**We add a sentence clarifying this to section 2.3**

5. P3, l12. I think there is an error in the projection details provided. The std lat is most likely -71 degs and central meridian will be 0 degs not 71 degs. Otherwise it's all rotated with a non std pll.

**Corrected.**

6. P1, l23. Wrong reference to Bamber 2012. Should be Bamber, J. L., Gomez Dans, J. L., and Griggs, J. A. (2009), A new 1 km digital elevation model of the Antarctic derived from combined satellite radar and laser data. Part I: Data and methods, The Cryosphere 3(2), 101-111. Not the NSIDC URL.

**Corrected.**

7. P4, l31-35. The text in brackets could be better phrased. It's not picking a travel time but picking a point on the leading edge of the waveform that represents the surface. This point is a function of the retracking procedure. With a threshold retracker, the bias is a function of the choice of threshold. If the bias is really due to penetration (=> using a threshold that picks a point below the surface) then this will be a function of snowpack properties and, in particular, density. This may not have a clear relationship with elevation but should correlate with, say, surface density as estimated from an RCM. See, for example, Wang, F., Bamber, J. L., and Cheng, X. (2015), Accuracy and Performance of CryoSat-2 SARIn Mode Data Over Antarctica, Geoscience and Remote Sensing Letters, IEEE, PP(99), 1-5, doi:10.1109/LGRS.2015.2411434.

**We have changed the section in brackets to read: "Strips with both Cryosat-2 and ICESat-1 registration within the precision thresholds allow for an estimate of the biases in Cryosat-2 height estimates due to the penetration of microwaves into the snow and firn layer (i.e. the penetration depth), or biases due to the retracking algorithm (i.e. where the retracker identifies a point on the leading edge of the waveform that does not correspond perfectly to the surface)."**

**We also added text to the end of the paragraph, to read:**

**The mean difference between the two corrections is -0.39 ± 0.35 m. We expect the bias in the Cryosat-2 data to depend on surface density and surface slope (Wang and others, 2015), but we do not have a straightforward way of predicting the bias, and we did not find a clear spatial or elevational dependence of the CS2-ICESat differences. Therefore, we added a uniform value of 0.39 m to the Cryosat-2-registered heights, regardless of the location of the strips and the surface type.**

8. P5, l4. Don't think "elevational" is a real word. Replace with elevation-related.

**Changed as suggested.**

---

## Author Response (AR2)

**Response to Editor's Comments (13 Feb 2019).**

We thank the editor for is careful and constructive suggestions. We have addressed each of his comments and adopted the suggestions in the revised manuscript. Each of the editors suggestions are listed below, with author responses in **bold font**.

1.12 « nearly ». Provide % of coverage in brackets to illustrate what is meant by « nearly ».

**Added [95%]**

4.10 South of 88° south

**Added**

4.17 « too » instead of « to »

**Corrected**

4.18 maybe I missed the info, but if not tell when/on what criteria the 31 % remaining strips were excluded.

This is explained by the following sentence, which we moved one sentence up and edited to make this point clearer. This now reads:

"The remaining 31% of strips were not visually inspected because we switched from inspecting every strip to only inspecting strips needed for a single mosaic coverage part way through the quality control process. This resulted in fewer inspected strips for regions inspected after this change in procedure."

4.27. accuracy. Ref for the 10 cm ? Shuman et al., GRL, 2006 maybe ?

**Added the Shuman et al 2006 reference.**

4.33. point cloud locations. Of the reference (CS2/GLAS) altimetry data ? Clarify.

**Added: "then interpolated to the Cryosat-2 SARIn-mode point cloud locations."**

4.34. Linear trend to what ? The time series of the CS2 at this location? I do not feel the sentence is clear enough and this is an important aspect of your method.

Edited to read: "For Cryosat-2 registrations, we estimated the linear temporal trend in the surface height from the time-series of all points within each DEM, so that each altimetric

point measurement would provide an estimate of the surface height at the time of DEM acquisition."

5.4 A reference is needed to back up the moderate elevation change in the interior. An altimetry study. Maybe the recent Schroder et al. study in TC (https://www.the-cryosphere.net/13/427/2019/) ?

**We now reference Helm et al. 2014 "Elevation and elevation change of Greenland and Antarctica derived from CryoSat-2", with an "e.g." since many altimetry papers show this (such as those of Zwally, Pritchard, etc.)**

5.4 the term resolution is not clear. In the context of this paper, it makes us think about horizontal resolution. Do you mean rather vertical precision ?

**Edited to read: "where changes in surface elevation are expected to be less than the resolution of repeat surface height observations on sub-decadal timescales"**

5.6 Why is the altimetry point cloud « corrected » ? What correction did you apply to the altimetry data ?

**Removed – typo from previous revision.**

5.7-8. Why using different metrics for CS2 and ICESat? Not clear to the reader. Either use the same terminology to describe both corrections, or explain why different thresholds are applied. And in one case you stated that the DEMs are used for mosaicking and in the other case that the correction is applied. I assumed the correction is also applied to the DEMs compared to CS2...

**Edited to read: "For ICESat-1, we impose a lower maximum threshold in the standard deviation of the residuals of 0.35 m because such strips were mostly used over the flatter interior terrain of Cryosat-2's LRM coverage."**

5.12. « precision ». Is not it rather « accuracy » here ? As the mean bias is the main criteria. A thorough check of the terminology is required. Precision and accuracy are two different concepts (as you know I am sure).

**It's a bit ambiguous here, since the criteria are applied to the fit of registration (ie. the sigmas of the correction and its residuals). Therefore changed to : "within the bias correction uncertainty thresholds"**

We also replaced "precision" on page 8 line 15 with : "internal accuracy (i.e. between locations on a single DEM)"

5.15. « were negligible ». I would say « are assumed negligible ». You did not verify it but rather made this assumption. A reference is required here to back up this statement.

**Changed to "Such biases are assumed negligible for the 1064 nm wavelength pulse of ICESat-1's laser altimeter and..."**

5.20 how can you be sure this is a sensor related bias. Why not a real elevation change between the ICESat and the CS2 periods ? If you can discard this possibility, tell why and provide a reference.

Added to the previous sentence: "These strips were distributed across the entire area of Cryosat-2 SARIn coverage and, therefore, the mean difference between Cryosat-2 and ICESat-1 bias corrections should not be sensitive to local variability in surface elevation change between missions".

6.11 a reference is needed to back up this statement.

**As above, we add the reference to e.g. Helm et al. 2014,.**

6.18. double check the grammar of this sentence

**Corrected typo (edited "we masked as water")**

6.21 why using the 70th percentiles here while using 68th percentile elsewhere. Seems inconsistent.

**Corrected to be 68th, which are what these values are. 70th percentiles were used in an earlier version.**

6.27. « tend to cover » I think.

**Corrected.**

6.32. Here again a reference is needed.

**As above, we add the reference to e.g. Helm et al. 2014.**

8.18 Fig. X (?)

**Corrected to Fig. 9.**

Figure 6. Legend. Inconsistency between 68th percentile and LE70...

**Corrected**

Figure 2. The 1-to-1 line (dashed) would be a useful addition to the figure.

**1-to-1 line (dashed) added**

Supplement. It would be best to generate a single high resolution PDF including all the supplementary figures and their legends. Also the figures need to be annotated, especially the figures showing issues of ghosting and shadowing. Many readers will not know where the artefacts are in the images.

We have constructed this single pdf with all images and include annotations of artifacts discussed, although we also include the non-annotated versions. We have also revised the main text to more clearly reference the supplementary figures, using proper figure numbering (e.g. S1, S2, etc.). We have also included a paragraph on page 4 that more clearly describes the artifacts shown in the supplementary figures.

[revised manuscript text omitted]

**Moved (insertion) [1] Deleted: Deleted: Part way through the quality control process Deleted: Part way through the quality control process Deleted: Part way through the quality control process Deleted: resulting in a reduction in Deleted: the Deleted: number of Deleted: some r Deleted: se**

cloud locations. For Cryosat-2 registrations, we estimated the linear temporal trend in the surface height from the time-series of all points within each DEM, so that each altimetric point measurement would provide an estimate of the surface height at the time of DEM acquisition. We did not apply a similar time-dependent correction to the ICESat-1 data because the time span between the altimetry measurements and the DEM was much larger. Further, we only use ICESat-1 data in the absence of

5 quality Cryosat-2 SARIn mode data, which is predominantly in the slow-flowing interior of the ice sheet where changes in surface elevation are expected to be less than the resolution of repeat surface height observations on sub-decadal timescales (e.g. Helm et al. 2014).

The median difference between the DEMs and the altimeter point clouds provides an estimate of the DEM's vertical bias. For Cryosat-2 data, only vertical bias corrections with a 1-sigma uncertainty of less than 0.1 m and residuals with a standard

- 10 deviation of less than 1 m were used in mosaicking. For ICESat-1, we impose a lower maximum threshold in the standard deviation of the residuals of 0.35 m because such strips were mostly used over the flatter interior terrain of Cryosat-2's LRM coverage. A total of 6,679,897 km2 are covered by Cryosat-2 registered DEMs, or 29,901,958 km2 including repeat coverage (Fig. 1C), with registered ICESat DEMs covering 4,897,600 km2, including 8,739,128 km2 of repeat coverage (Fig. 1D). Strips with both Cryosat-2 and ICESat-1 registration within the bias correction uncertainty thresholds allow for an estimate of
- 15 the biases in Cryosat-2 height estimates due to the penetration of microwaves into the snow and firm layer (i.e. the penetration depth), or biases due to the retracking algorithm (i.e. where the retracker identifies a point on the leading edge of the waveform that does not correspond perfectly to the surface). Such biases are assumed pegligible for the 1064 nm wavelength pulse of ICESat-1's laser altimeter and, therefore, the difference between the ICESat-1 and Cryosat-2 bias corrections should give an estimate of the Cryosat-2 bias. Fig. 2 plots the vertical bias corrections from ICESat-1 and Cryosat-2 for 227 strips for which
- 20 standard deviations of residuals were less than 0.25 cm. These strips were distributed across the entire area of Cryosat-2 SARIn coverage and, therefore, the mean difference between Cryosat-2 and ICESat-1 bias corrections should not be sensitive to local variability in surface elevation change over the period between the two missions. The mean difference between the two corrections is  $-0.39 \pm 0.35$  m. We expect the bias in the Cryosat-2 data to depend on surface density and surface slope (Wang and others, 2015), but we do not have a straightforward way of predicting the bias, and we did not find a clear spatial or
- 25 elevational dependence of the CS2-ICESat differences, Therefore, we added a uniform value of 0.39 m to the Cryosat-2registered heights, regardless of the location of the strips and the surface type.

**3** Mosaicking**

Quality-controlled, strip DEMs were mosaicked into 100-km by 100-km tiles with a 1-km wide buffer on each side to enable coregistration and feathering between tiles. For each tile, strips with altimetry registration were added first, in order of ascending vertical error, with a linear distance-weighted edge feather applied to the strip boundaries. The error value at each pixel is the standard error from the residuals of the registration to altimetry, and the date stamp is the day of DEM acquisition. The ± 0.35 m errors in bias for Cryosat-2 registered tiles were not included in this error estimate. In areas where edges of strips have been feathered, the error and date stamp are averaged with the same weighting as the elevation. Once all registered strips

[revised manuscript text omitted]

×....